# Androgen Receptor Gene Pathway Upregulation and Radiation Resistance in Oligometastatic Prostate Cancer

**DOI:** 10.3390/ijms23094786

**Published:** 2022-04-26

**Authors:** Helen Saxby, Stergios Boussios, Christos Mikropoulos

**Affiliations:** 1Torbay & South Devon NHS Healthcare Foundation Trust, Lowes Bridge, Torquay TQ2 7AA, UK; h.saxby@nhs.net; 2Department of Medical Oncology, Medway NHS Foundation Trust, Windmill Road, Gillingham Kent ME7 5NY, UK; 3Faculty of Life Sciences & Medicine, School of Cancer & Pharmaceutical Sciences, King’s College London, London SE1 9RT, UK; 4AELIA Organization, 9th Km Thessaloniki–Thermi, 57001 Thessaloniki, Greece; 5St Lukes Cancer Centre, Royal Surrey County Hospital, Egerton Rd, Guildford GU2 7XX, UK; christos.mikropoulos@nhs.net

**Keywords:** prostate cancer, oligometastases, stereotactic ablative body radiotherapy, androgen deprivation therapy, androgen receptor

## Abstract

Stereotactic ablative body radiotherapy (SABR) is currently used as a salvage intervention for men with oligometastatic prostate cancer (PC), and increasingly so since the results of the Stereotactic Ablative Body Radiotherapy for the Comprehensive Treatment of Oligometastatic Cancers (SABR-COMET) trial reported a significant improvement in overall survival with SABR. The addition of androgen deprivation therapy (ADT) to localised prostate radiotherapy improves survival as it sensitises PC to radiotherapy-induced cell death. The importance of the androgen receptor (AR) gene pathway in the development of resistance to radiotherapy is well established. In this review paper, we will examine the data to determine how we can overcome the upregulation of the AR pathway and suggest a strategy for improving outcomes in men with oligometastatic hormone-sensitive PC.

## 1. Introduction

Prostate cancer (PC) is the second most commonly diagnosed cancer in males worldwide and the fifth most common cause of cancer-related death in men [1]. Despite diagnostic and therapeutic achievements, very few biomarkers have been implemented in routine clinical practice to date [2]. Angiogenesis plays a major role in the development and spread of PC [3]. Similar to other cancers, epigenetic and somatic or germline genetic modifications lead to a higher risk of PC and its progression [4]. At diagnosis, 80–90% of PCs are dependent on androgens. Androgen deprivation therapy (ADT), which reduces serum androgens and inhibits androgen receptors (ARs), is the cornerstone of PC treatment [5]. It is also one of the first examples of targeted anti-cancer therapy. Patients with high-risk non-metastatic PC who receive ADT with combination therapy (abiraterone and/or enzalutamide) have significantly better metastases-free survival and overall survival than those who receive ADT alone [6]. ARs are expressed in primary PC and metastases and regulate cell proliferation, apoptosis, migration, invasion and differentiation [7,8]. Several luteinising hormone-releasing hormone (LHRH) analogues are available that can effectively halt androgen production by the testicles and render men castrated within 2–4 weeks. The most commonly used agents are goserelin, leuprolide and triptorelin. A new addition to our armamentarium is LHRH antagonists, such as degarelix. LHRH antagonists achieve castration more quickly (within approximately 3 days) and avoid the initial surge in gonadotrophin or androgen levels seen with LHRH analogues, which can lead to the tumour flare effect. Despite being a very effective treatment initially, patients with metastatic PC become resistant to ADT in a median time of 12–18 months after the initiation of treatment [9].

Localised PC may be managed by radical prostatectomy, external beam radiotherapy or brachytherapy, or monitored within an active surveillance pathway (for patients with low-risk disease) or a watchful waiting pathway (for patients not suitable for potentially curative treatment options). Treatment depends on patient choice, fitness and the stage and grade of the PC. External beam radiotherapy is commonly used in patients with intermediate- or high-risk PC and can achieve high cure rates; the CHHIP trial reported a 90% failure-free survival at 5 years [10]. Despite this, many men relapse and require further treatment. Unfortunately, 30–50% of patients undergoing radical radiotherapy for localised prostate cancer will experience biochemical recurrence within 10 years [11]. Metastatic PC remains incurable despite much research and many therapeutic advances.

The oligometastatic paradigm was initially defined by Hellman and Weichselbaum in 1995. Oligometastatic disease is an intermediate stage of cancer between localised and systemic disease when a salvage approach is feasible. The likelihood of the oligometastatic state is dependent upon tumour biology. Tumours detected early in their progression may have metastases limited in number and anatomic site. Historically, oligometastatic disease has been treated with systemic therapies aimed at disease control. There is increasing evidence and interest in managing oligometastatic disease with ablative treatments [12]. The analysis of relapse patterns after the radical treatment of localised prostate cancer has indicated that most patients relapse with three or fewer metastases [13,14].

There is a significant discrepancy in the definition of oligometastases. The ESTRO-ASTRO consensus is that of 1–5 metastatic lesions which can all be safely treated. Oligo-recurrence is often used interchangeably with metachronous oligometastatic disease and can be defined as oligometastatic recurrence at least 3 months after the initial diagnosis. Oligoprogression is used to describe progression in a limited number of metastases, whilst the majority of metastases are stable or responding [15].

It is highly likely that an increasing number of patients will be diagnosed with oligometastatic PC due to advances in imaging, particularly with the increasing use of prostate specific membrane antigen (PSMA) positron emission tomography (PET) scans, which can detect disease recurrence at an earlier stage with lower prostate specific antigen (PSA) levels than conventional imaging (computed tomography and bone scintigraphy). The PSMA protein is a type 2 transmembrane protein expressed in large quantities on virtually all PC cells; its expression increases with the aggressiveness of the PC. PSMA is not expressed in high levels in normal prostate tissue. PSMA PET scans include a molecule linked to a radioisotope (often gallium-68) that specifically binds to the PSMA protein. PSMA is also a target for the treatment of PC [16,17,18].

In this review paper, we will examine the current data to determine how we can overcome the upregulation of the AR pathway and suggest a strategy for improving outcomes in men with oligometastatic hormone-sensitive PC treated with SABR. The possibility of further enhancing the response to radiotherapy with the use of androgen receptor-targeted agents will be highlighted.

## 2. Stereotactic Ablative Body Radiotherapy

Stereotactic ablative body radiotherapy (SABR) enables a very high dose of external beam radiotherapy to be delivered safely and with precision to a target volume. It is a non-invasive outpatient treatment. The doses delivered are biologically similar to those with brachytherapy and have equivalent or lower side effects [19]. SABR is a treatment option for patients who present with up to three or five sites of oligometastatic disease. These sites can include bone, lymph node and other soft tissue metastases. Treatment is usually delivered in three to five fractions on alternate days with a conventional linear accelerator, a magnetic resonance linear accelerator or a CyberKnife machine. A ‘day zero’ appointment is often scheduled where the patient attends the radiotherapy department for a preliminary run-through on the treatment machine to ensure that the safe delivery of treatment is going to be technically possible.

To date, multiple phase II randomised controlled trials have demonstrated the safety and efficacy of treating oligometastases with SABR. In the Stereotactic Ablative Radiotherapy for the Comprehensive Treatment of Oligometastatic Cancers (SABR-COMET) international randomised phase II trial, a median survival benefit of 22 months was noted for patients with 1–5 oligometastases treated with SABR compared to the palliative standard of care therapy across different tumour sites. This translated to an absolute survival benefit of 24.6% at 5 years. Importantly, SABR did not result in a detriment to the patients’ quality of life. Of the 99 patients randomised within this trial, 16 (16%) had PC. The trial did not report on the use of ADT with SABR in the patients within the intervention arm [20,21]. A meta-analysis of 21 trials comprising 943 patients and 1290 oligometastases found that SABR was safe and well tolerated in the oligometastatic setting with good rates of local control. A subgroup analysis was conducted for PC oligometastases, which had an excellent 1-year local control rate of 97.9% [22].

The Observation vs. SABR for Oligometastatic PC (ORIOLE) phase II trial randomised 54 men with hormone-sensitive PC and 1–3 oligometastases to receive SABR or observation. Patients did not receive ADT in either arm until evidence of disease progression. This trial reported an improvement in progression-free survival (PFS) with SABR with no additional grade three toxicities compared to the observation group. With a median follow up of 18.8 months, median PFS was not met for patients treated with SABR compared to 5.8 months for patients undergoing observation (*p* value = 0.002). The local control of oligometastases treated with SABR was excellent (98.9% at 6 months) [23].

In the Surveillance or Metastasis-directed Therapy for Oligometastatic Prostate Cancer Recurrence (STOMP) trial, asymptomatic patients with up to three oligometastatic sites of recurrent prostate cancer were randomised to either metastasis-directed therapy (MDT) of all lesions or surveillance. Of the 31 patients randomised to receive MDT, 25 patients (81%) received SABR, and the remaining six patients (19%) underwent surgery. Patients did not receive ADT with MDT. No grade 2 or higher toxicity was observed in the MDT arm. The primary end point was ADT-free survival. The 5-year ADT-free survival was 8% in the surveillance group and 34% in the MDT group [24,25].

SABR-COMET-3 and SABR-COMET-5 are phase III randomised controlled trials that are currently recruiting to further investigate the overall survival benefit of SABR for up to three and 4–10 oligometastases, respectively. The outcomes measured include overall survival, progression-free survival, toxicity, quality of life and an economic evaluation.

Despite very promising results from the SABR-COMET, ORIOLE and STOMP trials, there is space to further improve the outcome of patients with hormone-sensitive oligometastatic PC (Table 1). Following the paradigm of multiple large randomised controlled trials demonstrating a survival benefit with the addition of ADT to radiotherapy in localised PC, an obvious and relatively simple way to improve patient outcomes would be to combine ADT with SABR [26,27,28].

## 3. Androgen Deprivation Therapy in Localised Prostate Radiotherapy

Over 20 randomised controlled trials have demonstrated the importance of the addition of ADT to radiotherapy in PC [29]. It is generally recommended that patients with intermediate-risk PC receive 6 months of ADT and patients with high-risk PC receive a longer course of ADT (2–3 years); in addition, ADT should start 2 months before radical radiotherapy.

In 1997, Bolla et al. randomised patients with locally advanced prostate cancer to either radical prostate radiotherapy with 3 years of a LHRHa starting on the first day of radiotherapy or radiotherapy alone. Patients who received a LHRHa experienced better local control and an improved overall survival. D’Amico et al. also demonstrated a significant overall survival benefit with the addition of 6 months of ADT to radical radiotherapy in patients with intermediate- or high-risk prostate cancer. ADT commenced 2 months prior to radiotherapy, concurrently and adjuvantly [26,27,28].

The addition of ADT to radiotherapy has demonstrated a clear benefit in locally advanced prostate cancer. Several trials were subsequently conducted to investigate and determine the optimal duration of ADT with radiotherapy for PC. The TROG 96.01 trial investigated whether 3 months or 6 months of neoadjuvant ADT decreased clinical progression and mortality after radiotherapy for locally advanced prostate cancer. ADT began 2 months before RT in the 3-month duration ADT group and 5 months before radiotherapy in the 6-month ADT group. Six months of neoadjuvant ADT significantly decreased distant progression and all-cause mortality compared with radiotherapy alone. Three months of neoadjuvant ADT did not demonstrate the same benefit [30]. The EORTC trial published in 2009 randomised men with locally advanced prostate cancer to receive either 6 months of ADT or 3 years of ADT with radical radiotherapy. ADT consisted of LHRHa, which was commenced on the first day of radiotherapy. At 5 years, the overall mortality was 3.8% higher in the short-term ADT arm than the long-term ADT arm of the trial; however, this did not demonstrate statistical significance [31]. The RTOG 9202 trial compared 4 months of ADT with 2 years of ADT in locally advanced prostate cancer. ADT commenced 2 months before the start of radiotherapy. Two years of ADT resulted in a significant improvement in disease-free survival. The benefit of the longer course of ADT was greater in patients with a higher Gleason score (8 to 10), and a significant improvement in overall survival was also observed in this group of patients [32].

The timing of ADT in relation to radiotherapy is crucial in optimising the additional benefit. Simply increasing the duration of ADT in unselected patients does not necessarily improve outcomes [33]. As ADT is associated with adverse effects on the cardiovascular system, bone health and overall quality of life, it is imperative to treat only the patients who will benefit from ADT at the correct time and for the shortest duration indicated [33,34].

## 4. Radioresistance

Radioresistance is a major factor causing the failure of radiotherapy and poorer outcomes in patients. The mechanisms behind radioresistance in cancer cells are associated with a number of different biological processes, including hypoxia, autophagy and DNA damage repair [35]. Radiotherapy achieves its therapeutic effect by inducing DNA damage in cancer cells either directly or indirectly (via the production of free radicals) and cell death thereafter. Radiation-induced double-strand breaks are the most lethal type of DNA damage and may be repaired by one of two pathways: nonhomologous end joining (NHEJ) or homologous recombination. Among the poly (ADP-ribose) polymerase (PARP) inhibitors, veliparib is the one that mostly potentiates the effect of fractionated radiation through its impairment of double-strand break repair pathways [36].

DNA damage response (DDR) mechanisms exist which provide an important line of defence and may enable cells to repair the damage caused by radiotherapy and survive [37]. It is already known that the upregulation of DNA repair genes involved in DDR is associated with an increased risk of metastasis across multiple tumour types, and cancer cells that have acquired radioresistance have an enhanced DNA repair capacity [38,39]. DDR mechanisms sense DNA damage and subsequently activate repair pathways. Among germline and somatic mutations in PC, DDR defects represent 25% of them—of these, *BRCA* mutations are the most frequent mutation to occur [40]. The advent of PARP inhibitors in metastatic castration-resistant PC has popularized germline testing of mutations in DNA repair genes, such as *BRCA1* and *BRCA2*. Guidelines for the above have been framed as a necessity in the era of precision oncology [41]. DNA-dependent protein kinase catalytic subunits (DNAPKcs) are key components in the NHEJ pathway. Increased levels of DNAPKcs are linked to recurrence after radiotherapy and disease progression [37,42,43].

## 5. AR Signalling and Radiotherapy

Androgens acting through ARs are essential for normal prostate development and function [44]. The AR is a member of the steroid hormone receptor family of molecules found intracellularly in various tissues, including PC and metastases. In the testes and prostate in males, 5-alpha-reductase converts testosterone to the more potent androgen dihydrotestosterone (DHT). DHT binds to ARs which then translocate from the cytoplasm to the nucleus, where they bind to the androgen response element (ARE) and initiate a transcription response. ARs regulate cell proliferation, apoptosis, migration, invasion and differentiation [8,45]. High AR expression in PC has been shown to be associated with lower recurrence-free survival and disease progression [9,46]. There is heterogeneity in AR transcription levels amongst PC specimens, which could result in variability in ADT sensitivity. The AR is known to play an important role in DNA repair pathway activation [47,48].

There is significant variability in the response to radiotherapy among PCs. Several trials have attempted to offer a molecular background into PC relapse and radioresistance. Multiple randomised controlled trials have demonstrated a disease-free and overall survival benefit with the addition of ADT to radiotherapy in localised PC (Table 2); however, there is an incomplete understanding of the mechanism behind this [49,50].

Goodwin et al. demonstrated that ADT enhances the response (i.e., reduces cell survival and cell doubling time) of hormone-sensitive PC to radiotherapy in vitro [37]. The team went on to investigate the mechanisms behind radioresistance in PC and discovered that DNA damage induces AR activity, which subsequently promotes double-strand DNA repair and enhances resistance to DNA damage in vitro and in vivo. DNAPKcs initiate NHEJ DNA double-strand repair and are essential in AR-mediated DNA repair. ARs induce DNAPKcs expression and activity, and DNAPKcs enhance AR activity in a positive feedback circuit [37]. Polkinghorn et al. further investigated the mechanism by which ADT increases the response of PC to radiotherapy using in vitro and xenograft models [47]. Their data established that AR signalling increases the expression of DNA repair genes and enhances PC radioresistance by accelerating the repair of radiation-induced DNA damage. These studies highlight the importance of AR signalling in repairing DNA breaks caused by radiotherapy.

Spratt et al. provided evidence that AR expression and activity is durably upregulated following prostate radiotherapy in vitro and in vivo, and the degree of AR upregulation is correlated with survival in vitro [51]. Furthermore, the same team examined AR-related protein levels (including human glandular kallikrein 2 (hK2)) in a cohort of 227 men with intermediate-risk PC who did not receive adjuvant ADT. They documented an increase in 20% of cases in measurable hK2 protein after radiotherapy. In addition, men experiencing a rise in serum hK2 levels post radiotherapy were three times more likely to experience biochemical failure compared to men with no hK2 rise [51].

Nestler et al. set out to identify clinical markers to determine patients at high risk for radioresistance [52]. They performed tissue microarray testing on salvage prostatectomy and primary prostatectomy specimens and stained for 15 proteins that were suggested to be associated with radioresistance by in vitro findings. The team identified two protein biomarkers associated with relapse post radical radiotherapy: AR and aldo-keto reductase family 1 member C3 (AKR1C3). In this study, AR and AKR1C3 appeared to be the most promising biomarkers for radioresistance in PC [52].

These results all support the use of ADT administered alongside radiotherapy to improve patient outcomes by inhibiting AR activity (Figure 1).

## 6. Timing of ADT with SABR

The localisation of the target volume is essential for the planning and delivery of SABR to treat oligometastatic disease. The radiation oncologist must be able to visualise the oligometastasis to delineate it on the radiotherapy planning scan. It also needs to be visible on CT scans taken by the radiographers on the linear accelerators prior to, during and after treatment to ensure the accurate and safe delivery of SABR. ADT can result in a rapid reduction in the volume of soft tissue target lesions, which impairs the ability to delineate and safely treat. A possible way to overcome this problem would be to initiate ADT on day zero of treatment, using a newer LHRH antagonist (such as degarelix). This strategy would enable us to gain the benefit of ADT combined with radiotherapy without compromising dose coverage and the safe delivery of SABR to the target lesions.

As highlighted by Spratt et al., there is a correlation between a rise in AR-related protein levels following radiotherapy and treatment failure [51]. The addition of ADT in the adjuvant setting should cover for AR expression upregulation and reduce the emergence of radioresistance. The optimum duration of adjuvant ADT is unknown; a short course of up to 6 months should be sufficient to cover the initial AR upregulation. A short course of ADT is preferred to reduce the duration of ADT side effects and their impact on patients’ quality of life. Pisansky et al.’s phase III trial failed to demonstrate a benefit for prolonged neoadjuvant ADT with radiotherapy for localised PC, further suggesting the timing of ADT in relation to radiotherapy is more important than the duration of ADT [33].

## 7. Androgen Receptor-Targeted Agents

There is great potential to further improve the outcomes of radiotherapy in PC in both the localised and oligometastatic setting with the addition of other agents that inhibit ARs. Enzalutamide (formerly MDV3100) is a second-generation anti-androgen that is a potent antagonist of AR activity and its signalling pathway. It binds to the ligand-binding domain of the AR and is thought to function by various mechanisms that reduce the activity of the AR, including acting as an androgen antagonist, inhibiting nuclear import of the AR and blocking the binding of the AR to DNA. It displays a much higher affinity for ARs than other anti-androgens [53]. Enzalutamide improves overall survival in patients with hormone-sensitive and castration-resistant metastatic PC and is now commonly used in these settings [54,55]. Other AR targeted agents now licensed for the treatment of prostate cancer include daralutamide and apalutamide.

Preclinical trials have demonstrated a benefit of adding AR-targeting drugs in addition to radiotherapy in PC. In 2018 Ghashghaei et al.’s research demonstrated that enzalutamide potentiates the radiation response both alone and in combination with ADT in hormone-sensitive PC cells in vitro. Enzalutamide inhibits the activity of proteins (e.g., DNA-PKcs) which are involved in the DNA damage response and repair processing. The level of DNA-PKcs activity was significantly less in the cell lines treated with enzalutamide and radiotherapy compared to those treated only with radiotherapy 1 h after treatment. Enhanced apoptosis was also noted in the cells treated with enzalutamide. In addition, enzalutamide was found to be a much stronger radiosensitiser than ADT. A more effective and profound blockade of AR activity and signalling was noted with enzalutamide compared to ADT. Interestingly, the radiosensitising effect that enzalutamide had on the cell lines was greatly dependent on the dosing schedule. Enzalutamide administered 2 h before radiotherapy produced a significantly higher dose enhancement factor compared to administration 24 h before or 24 h after radiotherapy [56].

Enzalutamide may also be beneficial at improving radiotherapy outcomes in patients with metastatic castration-resistant PC. Sekhar et al. demonstrated that enzalutamide was able to radiosensitise both hormone-sensitive and hormone-resistant human PC cell lines and xenografts in mouse models. Enzalutamide inhibited DNA damage repair and enhanced radiation-induced tumour growth delay in both hormone-sensitive and hormone-resistant PC cells [57]. Ghashghaei et al. confirmed these findings in their research conducted in vitro, which was published in 2019 [58].

Further robust clinical research is required and should soon be readily available to confirm whether AR-targeting agents can potentiate the radiosensitising effects of ADT. Randomised phase III trials are investigating radiotherapy combined with apalutamide and 6 months of ADT (ATLAS (NCT03488810)), enzalutamide with 24 months of ADT (ENZARAD (NCT024464440)) and daralutamide with 96 weeks of ADT (DASL-HICaP (NCT041363530)) in localised prostate cancer. The DASL-HICaP trial is currently recruiting and aims to complete the enrolment of 1100 participants by January 2028. The ATLAS and ENZARAD trials have completed accrual but have yet to report any results [57].

## 8. Future Directions

It seems logical that the addition of AR-targeted drugs will further potentiate the radiosensitising effect of ADT; however, we await the results of several phase III randomised controlled trials to confirm this.

P53, ATM mutations and the loss of PTEN are known to be associated with human prostate cancer radiosensitivity; however, they have not yet demonstrated the ability to predict which prostate cancers will eventually fail post radiotherapy. An increased understanding of the cellular mechanisms that are important in intrinsic radioresistance gene expression profiling may eventually enable the development of personalised radiotherapy. Gene expression profiles could be used to identify which patients would benefit from radiosensitisation [57,58].

Another exciting and rapidly developing field is that of artificial intelligence (AI). AI can recognize complex patterns in medical data and provide quantitative results. It could potentially improve the efficiency and efficacy of radiotherapy [59].

## 9. Conclusions

SABR is an exciting and promising treatment option for men with oligometastatic PC; as such, its use and availability are rapidly increasing. It has been proven to be safe and effective at improving progression-free and overall survival in multiple randomised phase II trials, and several phase III trials are currently recruiting. It is an attractive option compared to other ablative strategies, as it is a non-invasive outpatient treatment that can be delivered on a conventional linear accelerator. Initially, it does require additional expertise amongst radiation oncologists, physicians and radiographers who are involved in treatment planning and delivery; however, it does not necessarily require additional equipment.

The overall survival benefit of adding ADT to radiotherapy is well documented in localised PC. To date, prospective randomised trials have not investigated the use of ADT alongside SABR. As such, there is no consensus as to the use and timing of ADT with SABR to treat hormone-sensitive oligometastatic PC. There is robust evidence indicating that AR activates DNA repair pathways, which provides a rationale behind the use of ADT with SABR for hormone-sensitive prostate oligometastases. We suggest using a LHRH antagonist (e.g., degarelix) for 6 months starting from day zero of SABR when treating oligometastatic disease in PC to optimise the response to radiotherapy and enable the safe localisation of the target for SABR planning and delivery. A short duration of ADT is advised to avoid the prolonged effects that ADT has on the cardiovascular system, bone health and quality of life.

## Figures and Tables

**Figure 1 ijms-23-04786-f001:**
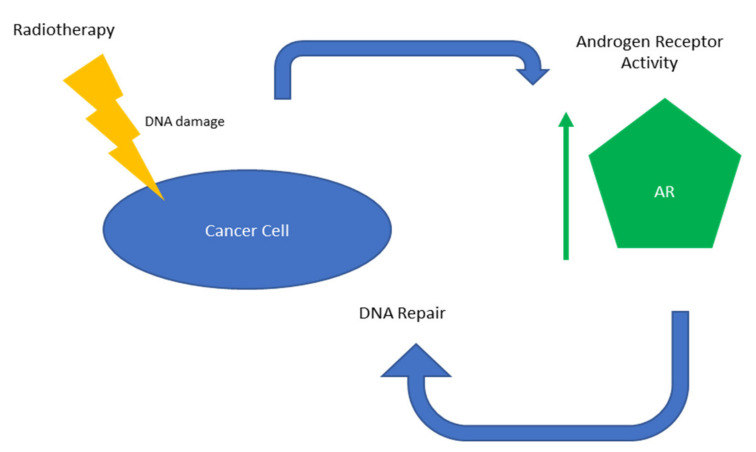
Overview of AR-mediated radioresistance.

**Table 1 ijms-23-04786-t001:** Radiotherapy for oligometastatic disease; summary of key trial findings.

Trial, Author (Year of Publication)	Summary and Main Findings	Reference
SABR-COMET, Palma, D.A.; et al. (2019)	Patients with 1–5 oligometastases were randomised to receive SABR or the palliative standard of care.	[20]
	Median survival benefit of 22 months and absolute survival benefit of 24.6% at 5 years with SABR.	
ORIOLE. Phillips, R.; et al. (2020)	Patients with PC and 1–3 oligometastases were randomised to receive SABR or observation.	[23]
	Improvement in PFS with SABR, with a median follow up of 18.8 months PFS not met in SABR arm versus 5.8 months in the observation arm.	
	Local control of oligometastases treated with SABR was 98.9% at 6 months.	
STOMP, Ost, P.; et al. (2018)	Patients with PC and 1–3 oligometastases were randomised to MDT or surveillance.	[25]
	5-year ADT-free survival was 34% in MDT arm and 8% in the surveillance arm.	

Abbreviations: ADT—androgen deprivation therapy; MDT—metastasis-directed therapy; PC—prostate cancer; PFS—progression-free survival; SABR—stereotactic ablative body radiotherapy.

**Table 2 ijms-23-04786-t002:** AR and radioresistance—summary of key trial findings.

Author (Year of Publication)	In Vivo or In Vitro	Main Findings	Reference
Goodwin, J.F.; et al. (2013)	Both	1. DNA damage induces AR activity. 2. ARs induce DNAPKcs expression and activity. 3. DNAPKcs enhance AR activity in a positive feedback loop.	[37]
Polkinghorn, W.R.; et al. (2013)	Both	ARs increase the expression of DNA repair genes.	[47]
Spratt, D.E.; et al. (2015)	Both	1. Radiotherapy upregulates AR expression and activity. 2. A rise in hK2 serum protein levels post radiotherapy are associated with an increased risk of biochemical relapse.	[51]
Nestler, T.; et al. (2019)	In vivo	AR and AKR1C3 are associated with PC relapse post radical radiotherapy.	[52]

Abbreviations: AR—androgen receptor; DNAPKcs—DNA-dependent protein kinase catalytic subunit; AKR1C3—aldo-keto reductase family 1 member C3; PC—prostate cancer; hK2—human glandular kallikrein 2.

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
