# Peer review of "Androgen Receptor Gene Pathway Upregulation and Radiation Resistance in Oligometastatic Prostate Cancer"

_ijms, 2022, doi:10.3390/ijms23094786_

Round 1

Reviewer 1 Report

Interesting cut-off title of the Review Article. 
Although the evidence is not established still interesting to discuss. 

The author may strengthen two interesting Topics. 1. is Radiation against oligo Mets. 2. Mechanism of radiation-based anti-tumor effect through AR and resistance. 

Major 

  1. Please make a table for a summarized radiation trial against oligo Mets including the following trial of SABR. 

    HORRAD
    https://www.ncbi.nlm.nih.gov/pubmed/30266309

    STAMPEDE

    https://www.sciencedirect.com/science/article/pii/S0140673618324863?via%3Dihub

  2.  Also please mention the upper two trials in the text of the SABR section. 
  3.  Could you make the diagram or figure indicating the mechanism of the AR-mediated anti-tumor effect of radiation+ radiation resistance? 

Author Response

Reviewer 1:

Interesting cut-off title of the Review Article.

Although the evidence is not established still interesting to discuss.

The author may strengthen two interesting Topics. 1. is Radiation against oligo Mets. 2. Mechanism of radiation-based anti-tumor effect through AR and resistance.

Major

  1. Please make a table for a summarized radiation trial against oligo Mets including the following trial of SABR.

HORRAD

https://www.ncbi.nlm.nih.gov/pubmed/30266309 

STAMPEDE

https://www.sciencedirect.com/science/article/pii/S0140673618324863?via%3Dihub

Thank you for your comments. I have included the table summarizing the SABR trials.

  1. Also please mention the upper two trials in the text of the SABR section.

Thank you very much for your comments. I haven’t included the STAMPEDE or HORRAD trials as these are not investigating the use of SABR. The STAMPEDE trial is investigating the benefit of high dose palliative radiotherapy to the prostate in patients with low volume metastatic prostate cancer. The HORRAD trial was not delivering SABR (radiotherapy was delivered in 2-3Gy/1#) to and patients could have any number of bony metastases (not oligometastases).

  1. Could you make the diagram or figure indicating the mechanism of the AR-mediated anti-tumor effect of radiation+ radiation resistance?

Thank you, a figure has been included giving an overview of AR-mediated radioresistance

Reviewer 2 Report

The authors have presented a paper about "Androgen Receptor Gene Pathway Upregulation and Radiation Resistance in Oligometastatic Prostate Cancer".

The topic is absolutely interesting and matter of debate within the clinical and scientific community.

I have a few cooments which I would like the authors to address:

1) In the introduction section it would be necessary to define more clearly the currently accepted defintion of oligometastases, oligorecurrences and oligoprogression to make the article more comprehensible to a larger audience

2) In my opinion figure 1 could be removed because it is of low quality and doen not add anything more to the article

3) The section "Androgen Deprivation Therapy in Localised Prostate Radiotherapy" is placed within the article for "educational" reason but could be in my view shortened for the purposes of the present article

4)  In the future directions section I recommend to include some referral to the possible role of Aritficial Intelligence and and new treatment straetegies (see PMID 34919226 for further details)

5) I believe that it could be better to split the "Conclusions and future directions" section placing the future direction before the final conclusions.

Author Response

Reviewer 2:

The authors have presented a paper about "Androgen Receptor Gene Pathway Upregulation and Radiation Resistance in Oligometastatic Prostate Cancer".

The topic is absolutely interesting and matter of debate within the clinical and scientific community.

I have a few cooments which I would like the authors to address:

1) In the introduction section it would be necessary to define more clearly the currently accepted defintion of oligometastases, oligorecurrences and oligoprogression to make the article more comprehensible to a larger audience

Thank you for your recommendation, these definitions have been added to the introduction.

2) In my opinion figure 1 could be removed because it is of low quality and doen not add anything more to the article

Figure 1 has been removed

3) The section "Androgen Deprivation Therapy in Localised Prostate Radiotherapy" is placed within the article for "educational" reason but could be in my view shortened for the purposes of the present article

This section has been shortened, thank you

4) In the future directions section I recommend to include some referral to the possible role of Aritficial Intelligence and new treatment strategies (see PMID 34919226 for further details)

Thank you for this recommendation , AI has now been referred to in the future direction section.

5) I believe that it could be better to split the "Conclusions and future directions" section placing the future direction before the final conclusions.

These amendments have been made, thank you for your comments

Round 2

Reviewer 1 Report

I do not see any figure for the following point but just a Table. 

The author will need to show the figure representing the mechanism of Radioresistance since this is not a common mechanism. 

  1. Could you make the diagram or figure indicating the mechanism of the AR-mediated anti-tumor effect of radiation+ radiation resistance?

Thank you, a figure has been included giving an overview of AR-mediated radioresistance

Author Response

Dear Reviewer,

I am deeply sorry that mistakenly, I did not upload the final version of the revised manuscript in the previous round. The figure was already created.

Apologies for the inconvenience caused.

Thank you for your time and your constructive comments.

Reviewer 2 Report

Accept

Author Response

I am deeply grateful for your time and your positive feedback.

Round 3

Reviewer 1 Report

Appropriate Correction has been made. No further request.